# Relational Multi-Task Learning: Modeling Relations between Data and Tasks

**Kaidi Cao**[*]        **Jiaxuan You**[*]        **Jure Leskovec**

Department of Computer Science, Stanford University
`{kaidicao, jiaxuan, jure}@cs.stanford.edu`

## Abstract

A key assumption in multi-task learning is that at the inference time the multi-task model only has access to a given data point but not to the data point's labels from other tasks. This presents an opportunity to extend multi-task learning to utilize data point's labels from other auxiliary tasks, and this way improves performance on the new task. Here we introduce a novel *relational multi-task learning* setting where we leverage data point labels from auxiliary tasks to make more accurate predictions on the new task. We develop MetaLink, where our key innovation is to build a knowledge graph that connects data points and tasks and thus allows us to leverage labels from auxiliary tasks. The knowledge graph consists of two types of nodes: (1) data nodes, where node features are data embeddings computed by the neural network, and (2) task nodes, with the last layer's weights for each task as node features. The edges in this knowledge graph capture data-task relationships, and the edge label captures the label of a data point on a particular task. Under MetaLink, we reformulate the new task as a link label prediction problem between a data node and a task node. The MetaLink framework provides flexibility to model knowledge transfer from auxiliary task labels to the task of interest. We evaluate MetaLink on 6 benchmark datasets in both biochemical and vision domains. Experiments demonstrate that MetaLink can successfully utilize the relations among different tasks, outperforming the state-of-the-art methods under the proposed relational multi-task learning setting, with up to 27% improvement in ROC AUC.

## 1 Introduction

The general idea of learning from multiple tasks has been explored under different settings, including multi-task learning (Caruana, 1997), meta learning (Finn et al., 2017), and few-shot learning (Vinyals et al., 2016; Cao et al., 2020a;b). While these learning settings have inspired models that can utilize relationships among tasks (Chen et al., 2019; Zamir et al., 2018; Sener & Koltun, 2018; Lin et al., 2019; Ma et al., 2020), they are not able to capture the full complexity of real-world machine learning applications. Concretely, when learning from multiple tasks, current approaches assume that the test data points have no access to the labels from other tasks when making predictions on a new task. However, this assumption oversimplifies potential useful knowledge in many real-world applications.

For example, multi-task learning studies simultaneously learning multiple predictive tasks to exploit commonalities between the tasks. At the test time the multi-task model predicts labels of a given data point for the tasks of interests, e.g., predicting whether the chemical compound $\mathbf{x}$ is non-toxic. At the same time, one may also have access to the data point's labels for some other auxiliary tasks, e.g., the compound $\mathbf{x}$ has a positive result on two toxicology tests. Such auxiliary task labels could greatly help with improved predictions.

However, current deep learning architectures cannot model such knowledge transfer between auxiliary tasks/labels and the target tasks. Naively concatenating the known labels to the input features has its limitations, especially since such labels are sparsely available, and it is also unclear how to use the approach for new and unseen tasks. Another potential solution to model such flexible and conditional inference is through generative models (Dempster et al., 1977; Koller & Friedman, 2009). Although generative models are powerful, they are notoriously data-hungry, thus it is very difficult to construct and train a generative model for high dimensional data (Turhan & Bilge, 2018).

---

[*]The two first authors made equal contributions.

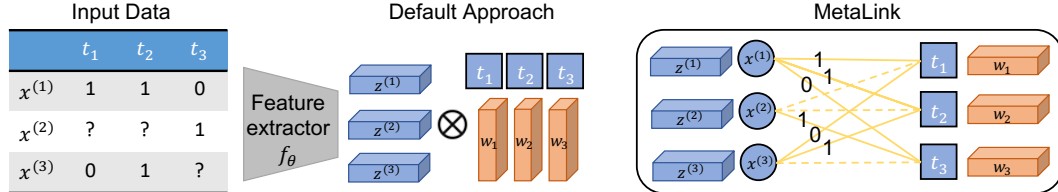

Figure 1: In the relational multi-task setting, the model learns to incorporate auxiliary knowledge in making predictions to achieve data efficiency. Concretely, given observations $\mathbf{x}^{(i)}$ and their labels $\{y_j^{(i)}\}$ (0/1 in this example) on subsets of tasks $\{t_j\}$, the goal is to build a model that can harness the auxiliary task labels $\{y_j^{(i)}\}$ and make predictions on a new task $t_n$. A standard approach is to build a multi-head deep neural network, with a prediction head for each individual task $t_j$. However, such approach cannot utilize auxiliary labels. In contrast, our proposed MetaLink reinterprets the last layer's weights of each task as task nodes and creates a knowledge graph where data points and tasks are nodes and labeled edges provide information about labels of data points on tasks. Then, when predicting data point's label for a given task $t_j$, MetaLink uses labels from other tasks to improve predictive performance.

Here we propose a new multi-task learning setting called *relational multi-task learning*. In our setting we distinguish between the target tasks (e.g., predicting molecule toxicity), which are tasks we aim to predict, and auxiliary tasks, which are tasks for which data point's labels are available at the inference time. Note that under our setting, each data point might have labels available for a different subset of auxiliary tasks. The goal then is to achieve strong predictive performance through leveraging labels of a given data point on some subset of auxiliary tasks.

To tackle the relational multi-task learning, we propose MetaLink[1], a general discriminative model that can explicitly incorporate the knowledge from auxiliary tasks. Our key innovation is to build a knowledge graph that connects different tasks $t_j$ and data points $\mathbf{x}^{(i)}$ (Figure 1). The first step of our approach is to take input data points $\mathbf{x}^{(i)}$ and the feature extractor (*i.e.*, neural network) $f_\theta$ to get its embedding $\mathbf{z}^{(i)}$. Then we build the knowledge graph which consists of two types of nodes: data nodes $\mathbf{x}^{(i)}$ and task nodes $t_j$. A data node $\mathbf{x}^{(i)}$ connects to a task node $t_j$ if data point $\mathbf{x}^{(i)}$ participates in task $t_j$ and the edge is annotated with the label $y_j^{(i)}$ of $\mathbf{x}^{(i)}$ on task $t_j$. We initialize data node features to be the last layer embedding $\mathbf{z}^{(i)}$ in the feature extractor $f_\theta$, and task node $t_j$ features are instantiated as the last layer's weights $\mathbf{w}_j$.

Given our knowledge graph, we reformulate the multi-task learning problem as a link-label prediction problem between data nodes and task nodes. This means that at the inference time MetaLink is able to use all the information about a given data point $\mathbf{x}^{(i)}$ (including its labels on auxiliary tasks $\{t_j\}$) to predict its label on a new task $t_n$. We solve this link label prediction learning task via a Graph Neural Network (GNN) (Hamilton et al., 2017; He et al., 2019; Xue et al., 2021). Unlike previous works, e.g., ML-GCN (Chen et al., 2019), that solely model relationships among tasks, MetaLink allows flexible and automatic modeling for data-task, data-data and task-task relationships.

We evaluate MetaLink on six benchmark datasets in both biochemical and vision domains under various settings. Empirical demonstrate that MetaLink can successfully utilize the relations among different tasks, outperforming the state-of-the-art methods under the proposed relational multi-task learning setting, with up to 27% improvement in ROC AUC.

## 2 RELATIONAL MULTI-TASK LEARNING SETTINGS

Here we first formally introduce the settings for relational multi-task learning. Suppose we have $m$ machine learning tasks $\{t_j\}_{j\in T}$, where $T = \{1, 2, ..., m\}$ is integers between 1 and m. We propose to categorize different settings in two dimensions: (1) whether the task is a relational task, *i.e.*, if auxiliary task labels can be used at inference time; and, (2) whether the task is a meta task, *i.e.*, if the

---

[1]Source code is available at https://github.com/snap-stanford/GraphGym

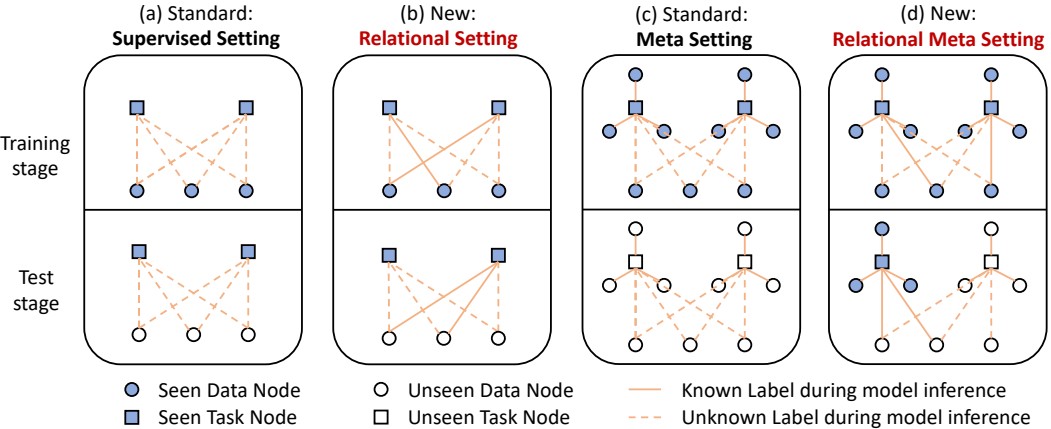

Figure 2: Our MetaLink framework allows for modeling four different multi-task learning settings: ◯ represent data nodes and □ represent task nodes. Blue represents the data/tasks seen in the training stage and white denotes the data/tasks seen only in the test stage. During model inference (for both the training and the test stage), the label of a data-task pair with solid line is known, while we want to predict labels of the data-task pairs with dotted lines.

task at test time has been seen at the training time. Altogether, there are four possible task settings, which are illustrated in Figure 2 and below.

**Standard supervised setting.** Let $\mathbf{x}^{(i)}$ denote the input and $y_j^{(i)}$ denote the corresponding label associated with task $t_j$, *i.e.*, $y_j^{(i)} \sim t_j$. Standard supervised multi-task learning can be represented as

$$\textbf{Train: } \left\{ \mathbf{x}^{(i)} \rightarrow \{y_j^{(i)} \sim t_j\}_{j \in T} \right\} \quad \textbf{Test: } \left\{ \mathbf{x}^{(i)} \rightarrow \{y_j^{(i)} \sim t_j\}_{j \in T} \right\}$$

where $\rightarrow$ connects the input and the output. *Training and test sets cover non-overlapping data points.* We use $y_j^{(i)}$ to concisely represent $y_j^{(i)} \sim t_j$ later on.

**Relational setting.** In relational setting, in addition to input $\mathbf{x}^{(i)}$, we assume we also have access to auxiliary task labels when making predictions. $T_{\text{aux}}$ and $T_{\text{test}}$ are partitions of integers ($T$) that relates to subsets of tasks. Specifically, $T_{\text{aux}}$ refers to the indices of tasks that input $\mathbf{x}$ has access to, and $T_{\text{test}}$ are the indices of tasks that we wish to predict; these two sets are non-overlapping, *i.e.*, $T_{\text{aux}} \cap T_{\text{test}} = \emptyset$, and input-dependent, *i.e.*, the partition $T_{\text{test}}^{(1)}$ and $T_{\text{test}}^{(2)}$ can be different. The input-output pairs are now in the form of

$$\textbf{Train: } \left\{ (\mathbf{x}^{(i)}, \{y_j^{(i)}\}_{j \in T_{\text{aux}}^{(i)}}) \rightarrow \{y_j^{(i)}\}_{j \in T_{\text{test}}^{(i)}} \right\} \quad \textbf{Test: } \left\{ (\mathbf{x}^{(i)}, \{y_j^{(i)}\}_{j \in T_{\text{aux}}^{(i)}}) \rightarrow \{y_j^{(i)}\}_{j \in T_{\text{test}}^{(i)}} \right\}$$

**Meta setting.** In the meta setting, we want to learn to predict unseen tasks at the test time. Formally, let $T_{\text{s}}, T_{\text{u}}$ denote the set of partitions for seen tasks (used at training time) and unseen tasks (used at test time), where $T_{\text{s}} \cap T_{\text{u}} = \emptyset$. Here, we have access to a batch of samples with labels as the support set $S$ and the objective is to correctly predict samples in the query set $Q$.

$$\textbf{Train: given } S = \left\{ (\mathbf{x}^{(i)}, \{y_j^{(i)}\}_{j \in T_{\text{s}}}) \right\}, \text{ predict } Q = \left\{ \mathbf{x}^{(i)} \rightarrow \{y_j^{(i)}\}_{j \in T_{\text{s}}} \right\}$$

$$\textbf{Test: given } S = \left\{ (\mathbf{x}^{(i)}, \{y_j^{(i)}\}_{j \in T_{\text{u}}}) \right\}, \text{ predict } Q = \left\{ \mathbf{x}^{(i)} \rightarrow \{y_j^{(i)}\}_{j \in T_{\text{u}}} \right\}$$

**Relational meta setting.** Relational meta setting combines the features of relational setting and meta setting. Similar to the meta setting, we aim to predict unseen tasks $T_{\text{u}}$ at test time; meanwhile, similar to the relational setting, we also assume having labels on a limited number of auxiliary tasks $T_{\text{aux}} \subseteq T_{\text{s}}$ to harness. Formally, we have a support set and a query set in the form of

$$\textbf{Train: given } S = \left\{ (\mathbf{x}^{(i)}, \{y_j^{(i)}\}_{j \in T_{\text{s}}^{(i)}}) \right\}, \text{ predict } Q = \left\{ (\mathbf{x}^{(i)}, \{y_j^{(i)}\}_{j \in T_{\text{aux}}^{(i)}}) \rightarrow \{y_j^{(i)}\}_{j \in T_{\text{s}}^{(i)} \setminus T_{\text{aux}}^{(i)}} \right\}$$

$$\textbf{Test: given } S = \left\{ (\mathbf{x}^{(i)}, \{y_j^{(i)}\}_{j \in T_{\text{u}}^{(i)}}) \right\}, \text{ predict } Q = \left\{ (\mathbf{x}^{(i)}, \{y_j^{(i)}\}_{j \in T_{\text{aux}}^{(i)}}) \rightarrow \{y_j^{(i)}\}_{j \in T_{\text{u}}^{(i)}} \right\}$$

## 3 METALINK FRAMEWORK

Next, we describe our MetaLink framework, which allows us to formulate the above multi-task learning settings in a single framework. In particular, MetaLink formulates them as a link label prediction task on a heterogeneous knowledge graph, this way, MetaLink can harness the relational information about data and tasks.

### 3.1 BUILD A KNOWLEDGE GRAPH ON TASK HEADS

We first recap the general formulation of a neural network. Given data points $\{\mathbf{x}^{(i)}\}_{i=1}^n$ and labels $\{\{y_j^{(i)}\}_{j \in T}\}_{i=1}^n$, a deep learning model can be formulated as a parameterized embedding function $f_\theta$ (which is deep) and a task head $f_\mathbf{w}$, consisting of only a single weight matrix $\mathbf{w}$. $f_\theta$ maps a data point $\mathbf{x}^{(i)}$ to a vector embedding space, $f_\theta(\mathbf{x}^{(i)}) = \mathbf{z}^{(i)} \in \mathbb{R}^D$. $f_\mathbf{w}$ then maps an embedding $\mathbf{z}^{(i)}$ to prediction $\hat{y}^{(i)} \in \mathbb{R}^2$, $f_\mathbf{w}(\mathbf{z}^{(i)}) = \mathbf{w}^T \mathbf{z}^{(i)} = \hat{y}^{(i)}$. When a task head involves multi-layer transformation, we have $f_\mathbf{w}(\mathbf{z}^{(i)}) = \mathbf{w}^T g(\mathbf{z}^{(i)}) = \hat{y}^{(i)}$, where $g(\cdot)$ can be an arbitrary function. In multi-task learning settings, people usually assign multiple task heads to a neural network. Suppose we have $m$ tasks $\{t_j\}_{j=1}^m$, then there are $m$ task heads such that $\hat{y}_j^{(i)} = f_{\mathbf{w}_j}(\mathbf{z}^{(i)})$.

Here, our observation is that the weights in task heads $\mathbf{w}_1, \ldots, \mathbf{w}_j$, and the data embeddings $\mathbf{z}^{(i)}$ play symmetric roles in a multi-task prediction task (due to the dot product). Therefore, instead of viewing weights $\mathbf{w}_1, \ldots, \mathbf{w}_j$ as parameters in a neural network, we propose to represent $\mathbf{w}_1, \ldots, \mathbf{w}_j$ as another type of input that supports the prediction. Essentially, we reformulate a task head from $\hat{y}_j^{(i)} = f_{\mathbf{w}_j}(\mathbf{z}^{(i)})$ to $\hat{y}_j^{(i)} = f_\phi(\mathbf{w}_j, \mathbf{z}^{(i)})$. This new perspective, where both task weights $\mathbf{w}_j$ and embedding $\mathbf{z}^{(i)}$ are viewed as input, enables us to build a more sophisticated predictive model $f_\phi$. $f_\phi$ in the sense that it contains two main steps, *i.e.*, GraphConv$(\cdot)$ and EdgePred$(\cdot)$. In general, EdgePred$(\cdot)$ has a similar model complexity as $f_{\mathbf{w}_j}$, whereas GraphConv$(\cdot)$ provides additional expressiveness.

In MetaLink, we propose to build a *knowledge graph* over task weights $\mathbf{w}_j$ and embedding $\mathbf{z}^{(i)}$. By building this knowledge graph, we can succinctly represent relationships between data points and tasks, as well as different multi-task learning settings. Concretely, a knowledge graph helps us easily express any data-task relationship (e.g., a data point has a label on a given task), data-data relationship (e.g., two data points are similar or not), or task-task relationship (e.g., hierarchy of different tasks). Moreover, a knowledge graph greatly simplifies the representation of all the multi-task learning settings that we outlined in Section 2; in fact, all the settings can be viewed as link label prediction tasks where different portions of the knowledge graph can be constructed, as illustrated in Figure 2.

We define the knowledge graph as $G = \{V, E\}$, where $V$ is the node set and $E \subseteq V \times V$ is the edge set. We define two types of nodes, data nodes $V_d = \{\mathbf{x}^{(1)}, \ldots, \mathbf{x}^{(n)}\}$ and task nodes $V_t = \{t_1, \ldots, t_m\}$. We can then define edges between data and task nodes $E_{dt} \subseteq V_d \times V_t$, within data nodes $E_{dd} \subseteq V_d \times V_d$, and within task nodes $E_{tt} \subseteq V_t \times V_t$. MetaLink framework can work with all three types of edges; however, since most benchmark datasets do not have information on data-data or task-task relationship, we focus on data-task relationship $E_{dt}$ in the remaining discussions. Specifically, we define $E_{dt}$ based on task labels, $E_{dt} = \{(\mathbf{x}^{(i)}, t_j) \sim y_{j \in T_{\text{aux}}}^{(i)}\}$, *i.e.*, we connect a data node $\mathbf{x}^{(i)}$ with a task node $t_j$ if label $y_j^{(i)}$ exists.

### 3.2 LEARN FROM THE KNOWLEDGE GRAPH VIA A HETEROGENEOUS GNN

Given the constructed graph, we then discuss how MetaLink learns from the built knowledge graph.

**Initialize node/edge features.** First, we initialize features for the knowledge graph. Concretely, we initialize data node features to be the data embeddings $\mathbf{z}^{(i)}$ computed from the feature extractor $f_\theta(\cdot)$, $\mathbf{h}_i^{(0)} = \mathbf{z}^{(i)}$. We initialize the seen task node embeddings using the weights in task heads $\mathbf{w}_j$, $\mathbf{h}_j^{(0)} = \mathbf{w}_j$. In meta settings where unseen task nodes appear during test time, we will initialize those nodes by a constant vector of $\mathbf{1}$; such initialization ensures that MetaLink can generalize to unseen tasks, since the node feature construction process is inductive.

---

[2] Without loss of generality, we assume a task is a binary classification or 1-D regression

**Predict via a heterogeneous GNN.** We implement the predictive model $f_\phi$ over data nodes and task nodes as a GNN. The goal of GNN is to learn expressive node embeddings $\mathbf{h}_v$ based on an iterative aggregation of local network neighborhoods. The $l$-th iteration of GraphConv($\cdot$), or the $l$-th layer, can be written as:

$$\mathbf{h}_v^{(l)} = \text{AGG}^{(l)}\Big(\{\text{MSG}^{(l)}(\mathbf{h}_u^{(l-1)}), u \in \mathcal{N}_G(v)\}, \mathbf{h}_v^{(l)}\Big) \tag{1}$$

where $\mathbf{h}_v^{(l)}$ is the node embedding after $l$ iterations, $\mathbf{h}_v^{(0)}$ have been initialized as explained above, and $\mathcal{N}_G(v)$ denotes the direct neighbors of $v$. AGG is the acronym for aggregation function and MSG is the acronym for message function. We perform $L$ GNN layers on top of the knowledge graph $G$ that we have built. After updating data and task node embeddings, we can make predictions on the given task through EdgePred($\cdot$) in the form of $\hat{y}_j^{(i)} = \text{MLP}\Big(\text{CONCAT}(\mathbf{h}_i^{(L)}, \mathbf{h}_j^{(L)})\Big)$.

In general, MetaLink should work with any GNN architecture that follows the definition in Equation 1. We use the GraphSAGE layer (Hamilton et al., 2017) in MetaLink ($\mathbf{W}^{(l)}$, $\mathbf{U}^{(l)}$ are trainable):

$$\mathbf{h}_v^{(l)} = \mathbf{U}^{(l)}\text{CONCAT}\Big(\text{MEAN}\big(\{\text{RELU}(\mathbf{W}^{(l)}\mathbf{h}_u^{(l-1)}), u \in \mathcal{N}(v)\}\big), \mathbf{h}_v^{(l-1)}\Big) \tag{2}$$

Next, we discuss the special GNN design used in MetaLink which has shown to be successful.

**Special GNN designs in MetaLink.** We make three extensions in the formulation in Equation 1. First, since there are two types of nodes in our knowledge graph, we define different message passing functions for different message types, *i.e.*, the message from data nodes to task nodes, and the message from task nodes to data nodes. Second, we include edge features in the message computation. This is especially important for our formulation, since the task label values $y_j^{(i)}$ are included as edge features, and should be considered during GNN message passing. Concretely, we extend Equation 2 into:

$$\mathbf{h}_v^{(l)} = \mathbf{U}^{(l)}\text{CONCAT}\Big(\text{MEAN}\big(\{\text{RELU}(\mathbf{W}_{\mathbf{1}[v \in V_d, u \in V_t]}^{(l)}\mathbf{h}_u^{(l-1)} + \mathbf{O}^{(l)}y_v^{(u)}), u \in \mathcal{N}(v)\}\big), \mathbf{h}_v^{(l-1)}\Big) \tag{3}$$

where $\mathbf{1}[v \in V_d, u \in V_t]$ indicates the message type (whether from task to data, or data to task), and $\mathbf{O}^{(l)}$ is an additional trainable weight that allows task label $y_v^{(u)}$ participates in message passing. Finally, we let each GNN layer make a prediction and sum them up as the final prediction; this way, the final prediction is obtained from mixed information from different hops of node neighbors. We observe this multi-layer ensemble technique can help MetaLink make robust predictions.

**MetaLink for relational meta setting.** Here we provide a detailed description on how to apply MetaLink for the relational meta setting in Algorithm 1. At training time, since most of the existing multi-label datasets are not designed for meta setting, we manually simulate such setting by sampling a mini batch with support $S = \Big\{(\mathbf{x}^{(i)}, \{y_j^{(i)}\}_{j \in T_s^{(i)}})\Big\}$ and query set $\Big\{(\mathbf{x}^{(i)}, \{y_j^{(i)}\}_{j \in T_{\text{aux}}^{(i)}}) \rightarrow \{y_j^{(i)}\}_{j \in T_s^{(i)} \setminus T_{\text{aux}}^{(i)}}\Big\}$. We make sure the sampled meta tasks $T_s^{(i)}$ and auxiliary knowledge tasks $T_{\text{aux}}^{(i)}$ have no intersection. We first use the feature extractor $f_\theta$ to get data embeddings and use the embeddings to initialize data nodes. To initialize task nodes, we either: (1) use $\mathbf{1}$ to initialize if the task is a meta task; Or, (2) use trained weights otherwise. We construct the edge set $E$ by connecting each data-task pair based on $\Big\{\{y_j^{(i)}\}_{j \in T_s^{(i)}}\Big\}$ in the support set and $\Big\{\{y_j^{(i)}\}_{j \in T_{\text{aux}}^{(i)}}\Big\}$ (we eliminate the relations we want to predict) in query set. Now that we have all the components of the knowledge graph, we can apply the predictive graph model $f_\phi$ to learn expressive data and task node embeddings and make predictions.

At test time, we use the same pipeline as in training to construct the knowledge graph and run inference. Please refer to Appendix A for the pipelines of relational or meta settings.

## 4 EXPERIMENTS

Here we experimentally show that our proposed MetaLink flexibly handles different settings and leverages knowledge from auxiliary tasks. We first evaluate our algorithms on Tox21 (Huang et al., 2016), Sider (Kuhn et al., 2016), ToxCast (Richard et al., 2016), and MS-COCO (Lin et al.,

---

**Algorithm 1** MetaLink Training in Relational Meta Setting

---

**Require:** Dataset $\mathcal{D}_{\text{train}} = \{(\mathbf{x}, y)\}$. A parameterized embedding function $f_\theta$. Last layer weights for each task $\mathbf{w}_j$. A parameterized heterogeneous GNN $f_\phi$. Number of GNN layers $L$.
1: **for** each iteration **do**
2:     $S, Q \leftarrow \text{SampleMiniBatch}(\mathcal{D}_{\text{train}})$                        ▷ Simulate meta setting in training
3:     $\{\mathbf{z}\} \leftarrow f_\theta(\mathbf{x})$ for $\mathbf{x} \in (S, Q)$
4:     $V_d^{(0)} = \{\mathbf{h}_i^{(0)} \leftarrow \mathbf{z} \text{ for } \mathbf{z} \in \{\mathbf{z}\}\}$                    ▷ Initialize data nodes
5:     $V_t^{(0)} = \{\mathbf{h}_j^{(0)} \leftarrow \mathbf{1} \text{ if meta else } \mathbf{w}_j \text{ for each } \mathbf{w}_j\}$      ▷ Initialize task nodes
6:     $E = \{\mathbf{e}_{ij} \leftarrow (\mathbf{x}^{(i)}, t_j) \text{ for } y_j^{(i)} \in (S, Q)\}$            ▷ Initialize edges
7:     **for** $l = 1$ to $L$ **do**
8:         $V_d^{(l)}, V_t^{(l)} \leftarrow \text{GraphConv}(V_d^{(l-1)}, V_t^{(l-1)}, E)$ with $f_\phi$
9:     $\text{logits} \leftarrow \text{EdgePred}(V_d^{(L)}, V_t^{(L)})$ with $f_\phi$
10:    $\text{Backward}\left(\text{Criterion}(\text{logits}, \{\{y_j^{(i)}\}_{j \in T_s^{(i)} \setminus T_{\text{aux}}^{(i)}}\} \in Q)\right)$

---

2014) datasets with various controllable settings on relational multi-task learning. To additionally demonstrate the advantage of MetaLink, we also include experiments on a well-studied task: few-shot learning. Our core algorithm is developed using PyTorch (Paszke et al., 2017). We use one NVIDIA RTX 8000 GPU for each experiment and the most time-consuming one (MS-COCO) takes less than 24 hours. Please refer to Appendix A for additional low-level implementation details.

## 4.1 EVALUATING METALINK ON RELATIONAL MULTI-TASK SETTINGS

**Datasets.** We simulate four relational multi-task settings (Figure 2) using four widely-used multi-label datasets. Tox21 (Huang et al., 2016) contains 12 different toxicological experiments for each sample with binary labels (active/inactive). Sider (Kuhn et al., 2016) is a database of marketed drugs and adverse drug reactions (ADR), grouped into 27 tasks. ToxCast (Richard et al., 2016) contains about 8K pairs of molecular graphs and corresponding 617-dimensional binary vectors that represent different experimental results. Microsoft COCO (Common Objects in Context) (Lin et al., 2014) is originally a large-scale object detection, segmentation dataset. By counting whether each type of object exists in a scene as a single task, it also serves as the default large-scale dataset for benchmarking multi-label classification in vision. There are 80 binary classification tasks with an average of 2.9 positive labels per image.

**Experimental setting.** We evaluate MetaLink on all four settings described in Figure 2, and we summarize the detailed configurations as follows. STANDARD SETTING: standard supervised learning. In order to have a fair comparison, we evaluate the same set of tasks with unknown labels as we sampled in relational settings. RELATIONAL SETTING: we assume each example has access to labels of 20% of all tasks in each dataset. We evaluate on the rest of the tasks with unknown labels. META SETTING: we hold out 20% of the tasks at the training time. We only evaluate on the held-out tasks at the test time. We use 256-shot setting, meaning at test time, we use 256 data points as a support set to initialize the prototypes of unseen tasks. The reason why the number of shots is much larger than what is commonly used in few-shot learning (1-shot, 5-shot) is that positive labels are sometimes sparse in certain tasks. RELATIONAL META SETTING: we hold out the same 20% of the tasks at the training time as in the meta setting. At test time, we assume each unseen task has access to 20% of the labels of seen tasks. We will release our splits of meta setting to promote reproducibility.

**Baselines.** Though the main motivation of our work is to utilize auxiliary task labels, we still include a few baselines under the standard supervised setting in order to benchmark state-of-the-art results. The simplest one is (1) Empirical risk minimization (ERM): we train the network with cross-entropy loss under the standard supervised setting; (2) Various commonly used graph neural network architectures designed for molecules, MPNN (Gilmer et al., 2017), DMPNN (Yang et al., 2019), MGCN (Lu et al., 2019), AttentiveFP (Xiong et al., 2019); (3) GROVER (Rong et al., 2020) integrates Message Passing Networks into the Transformer-style architecture. By leverage pretraining, it achieves state-of-the-art results on the aforementioned molecule datasets; (4) Baseline++ (Chen et al., 2018): since there is no prior work on addressing meta setting for multi-task learning, we adapt Baseline++ from few-shot

Table 1: Results of different multi-task learning settings on graph classification tasks, measured in ROC AUC. MetaLink can successfully utilize the relations among different tasks, outperforming the state-of-the-art method GROVER (Rong et al., 2020) under the relational setting.

| Method | Setting | Tox21 (12 tasks) | Sider (27 tasks) | ToxCast (617 tasks) |
|---|---|---|---|---|
| MPNN (Gilmer et al., 2017) | | 80.8±2.4 | 59.5±3.0 | 69.1±1.3 |
| DMPNN (Yang et al., 2019) | | 82.6±2.3 | 63.2±2.3 | 71.8±1.1 |
| MGCN (Lu et al., 2019) | Standard | 70.7±1.6 | 55.2±1.8 | 66.3±0.9 |
| AttentiveFP (Xiong et al., 2019) | | 80.7±2.0 | 60.5±6.0 | 57.9±1.0 |
| GROVER(48M) (Rong et al., 2020) | | 81.9±2.0 | 65.6±0.6 | 72.3±1.0 |
| GROVER(100M) (Rong et al., 2020) | | *83.1*±2.5 | *65.8*±2.3 | *73.7*±1.0 |
| **MetaLink** | Standard | 82.3±2.2 (KG layer = 1) | 60.9±2.4 (KG layer = 5) | 69.3±1.6 (KG layer = 3) |
| | **Relational** | **83.7±1.9** (KG layer = 1) | **76.8±3.0** (KG layer = 2) | **79.4±1.0** (KG layer = 4) |
| | Meta | 77.5±2.1 (KG layer = 2) | 57.9±5.0 (KG layer = 2) | 71.3±2.2 (KG layer = 2) |
| | **Relational**+Meta | **79.2±2.9** (KG layer = 2) | **65.4±4.3** (KG layer = 5) | **84.3±1.2** (KG layer = 5) |

learning to this setting. We first train a feature extractor on the training set. At test time we use the support set to train a linear classification layer.

**Results on biochemical datasets.** Table 5 summarizes the results on Tox21, Sider, and ToxCast datasets. We first clarify that when performing MetaLink under the standard setting, where there is no auxiliary task label to leverage, the knowledge graph we built degenerates to a trivial set since the edge set is empty. It has a negligible difference compared with training a vanilla network with cross-entropy loss. Thus, it is reasonable that MetaLink under the standard setting has similar performance with the baselines. GROVER has better performance because it leverages large-scale pretraining. In terms of comparing relational setting with standard setting, we conclude that the proposed MetaLink successfully leverages the auxiliary knowledge for each sample. Notably, with auxiliary labels from 20% of the tasks, MetaLink surpassed GROVER with large-scale pretraining by a large margin. In addition, we observe similar improvements between meta and relational meta settings. The empirical results for meta or relational meta setting are very close to the performance in standard setting, proving the potential of data-efficient research in biochemical domains.

**Results on MS-COCO.** Table 2 summarizes the results on MS-COCO. Note that as also mentioned in the settings, we need a large support set to ensure each task in the meta testing stage has at least one positive label. This blocks the feasibility of using a large model and input size as it is done in common benchmarks. Thus, we use ResNet-50 with an input size of 224 for all the experiments. We observe consistent improvements for both relational setting and relational meta setting. MetaLink also outperforms Baseline++ in the meta setting. Note that since label distribution for most of the tasks is highly skewed, mAP degrades more in the meta setting than for the biochemical datasets.

Table 2: Results on MS-COCO (80 tasks) dataset. We report the average accuracy and standard deviation over 5 runs on the validation set. We use ResNet-50 and our input size is 224. MetaLink achieves the best performance in relational settings through harnessing the auxiliary labels efficiently.

| Method | Setting | mAP |
|---|---|---|
| ML-GCN (Chen et al., 2019)[1] | Standard | 69.15 ± 0.19 |
| ERM | Standard | 71.22 ± 0.15 |
| Baseline++ (Chen et al., 2018) | Meta | 30.46 ± 0.69 |
| **MetaLink** | Standard | 71.58 ± 0.16 |
| | **Relational** | **75.36 ± 0.16** |
| | Meta | 41.75 ± 0.92 |
| | **Relational**+Meta | **49.73 ± 0.88** |

## 4.2 ABLATION STUDIES

**Does MetaLink truly learn to utilize correlations between tasks?** To better understand the improvement of our algorithm, we first plot the Pearson correlation heat map on Sider (Figure 3). We

---

[1]https://github.com/Megvii-Nanjing/ML-GCN

Figure 3: Pearson correlation heat map on 27 tasks of Sider dataset.

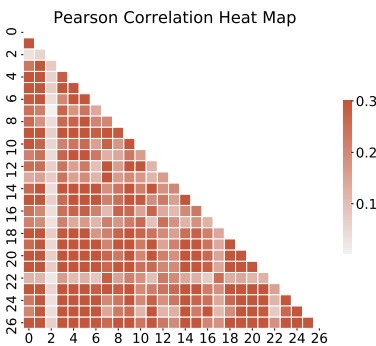

Table 3: ROC AOC on 3 tasks with the lowest average Pearson correlation to other tasks (tasks 2, 17, 22), as well as 3 tasks with the highest average Pearson correlation to other tasks (tasks 5, 20, 21) on the Sider dataset. We observe MetaLink gains more improvement on the tasks with high Pearson correlation.

| Setting | Low Corr Tasks | High Corr Tasks |
|---|---|---|
| Standard | $57.5 \pm 8.4$ | $61.0 \pm 4.1$ |
| **Relational** | **$70.1 \pm 7.8$** | **$80.6 \pm 3.3$** |
| Improvement | 21.9% | 32.1% |

then find the top 3 tasks with the highest and lowest average correlation with respect to the rest of the tasks. We report the performance of MetaLink on these two subsets of tasks, respectively (Table 3). We observe that MetaLink demonstrates larger improvement on tasks with higher correlations. This experiment verifies that MetaLink learns to utilize correlations between tasks as expected.

**How does MetaLink perform with varying ratio of auxiliary task labels?** We vary the ratio of additional labels per test point and report the results in Figure 4. We observe consistent improvement as we gradually increase the number of auxiliary labels in each dataset. This experiment demonstrates that MetaLink can successfully utilize marginal information whenever auxiliary task labels are added to the dataset. The improvement is usually prominent for the first few added labels.

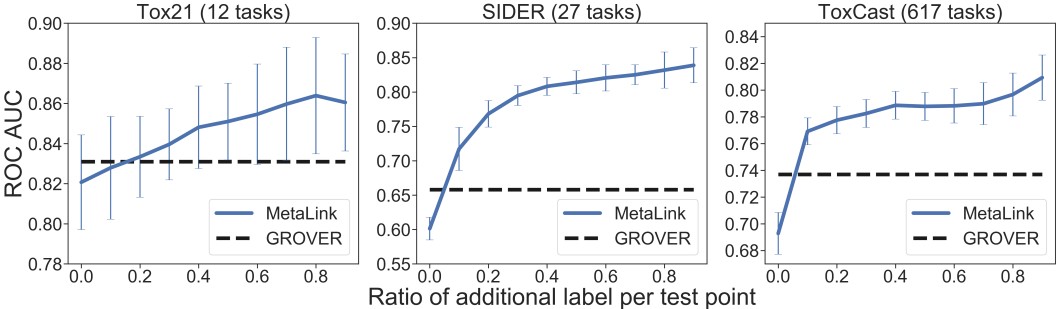

Figure 4: We vary the ratio of auxiliary labels per test point and plot the ROC AUC with the error bar. We also plot the state-of-the-art method (GROVER) as the dash line which cannot utilize additional auxiliary task labels. The performance of MetaLink consistently improves as more tasks are utilized. In Sider, MetaLink can outperform GROVER by up to 27% when 90% auxiliary labels are provided.

### 4.3 EVALUATING METALINK ON STANDARD FEW-SHOT LEARNING DATASETS

Since the relational multi-task setting is novel, there is a very small number of baselines that we can compare against. The major aim of this section is to show the advantage of MetaLink in a well-studied problem: few-shot learning. Note that there is a slight difference between the meta setting above and few-shot learning setting here. For few-shot learning, for all the link label prediction tasks related to an input, there will be only one positive link. This inductive bias could be easily incorporated by modeling all the link label predictions together using cross-entropy loss.

**Experimental setting.** We evaluate performance on two standard benchmarks: mini-ImageNet (Vinyals et al., 2016) and tiered-ImageNet (Ren et al., 2018). We compare against MatchNet (Vinyals et al., 2016), Baseline++ (Chen et al., 2018), MetaOptNet (Lee et al., 2019), and Meta-Baseline (Chen et al., 2020b), who only assume the input is a vector.

Table 4: Results on 5-way, 5-shot classification on mini-ImageNet and tiered-ImageNet datasets. We report the average accuracy and standard deviation over 800 randomly sampled episodes. MetaLink demonstrates consistent improvement from stacking KG layer = 0 to 2.

| Method | mini-ImageNet | tiered-ImageNet |
|---|---|---|
| MatchNet (Vinyals et al., 2016) | $78.72 \pm 0.15$ | $80.60 \pm 0.71$ |
| Baseline++ (Chen et al., 2018) | $77.76 \pm 0.17$ | $83.74 \pm 0.18$ |
| MetaOptNet (Lee et al., 2019) | $78.63 \pm 0.46$ | $81.56 \pm 0.53$ |
| Meta-Baseline (Chen et al., 2020b) (KG layer = 0) | $79.26 \pm 0.17$ | $83.29 \pm 0.18$ |
| **MetaLink** (KG layer = 1) | $79.86 \pm 0.18$ | $83.91 \pm 0.17$ |
| **MetaLink** (KG layer = 2) | $\mathbf{81.13 \pm 0.17}$ | $\mathbf{84.68 \pm 0.17}$ |

**Results.** Table 4 shows that our MetaLink outperforms the standard few-shot learning benchmarks. Note that if we set KG Layer = 0, the proposed MetaLink degenerates to Meta-Baseline. The experiments clearly demonstrate the benefits of building a knowledge graph on the last layer. Furthermore, as an ablation study we manipulate the number of KG layers and find that in the few-shot image recognition setting, there's an improvement for stacking 2 KG layers instead of 1, meaning non-linearity is useful. We do not observe further improvements for more than 3 layers.

## 5  RELATED WORK

**Multi-task learning.** Multi-task learning is a learning paradigm that jointly optimizes a set of tasks with shared parameters. It is generally believed that relations across different tasks can improve the overall performance. Some works cast it as a multi-objective optimization problem and introduce multiple gradient-based methods to reduce negative transfer among tasks (Fliege & Vaz, 2016; Lin et al., 2019). Other works assign (adaptive) weights for different tasks using certain heuristics (Kendall et al., 2018; Chen et al., 2018). Our empirical study is also closely related to multi-label learning, where the problem is usually decomposed into multiple binary classification tasks (Tsoumakas & Katakis, 2007). There is a line of work learning to leverage the relationship among tasks (Haller et al., 2021; Zamir et al., 2020). Wang et al. (2016) utilized recurrent neural networks to transform labels into embedded label vectors to learn the correlation among labels. The most recent work is ML-GCN (Chen et al., 2019), which uses a GCN to map label graph into a set of inter-dependent classifiers. Although the major motivation of our work is also about leveraging correlation among tasks, our problem formulation is new and thus yields a novel algorithm.

**Exploring graph structure for data and tasks.** There are prior works exploring graph structure for data points or tasks, and graph structure was proven to be effective in some tasks. Satorras & Estrach (2018) explores graph neural representations over *data points only* for few-shot learning. Besides, some works study how to transfer knowledge among tasks through constructing a graph over *task nodes/classifiers only* (Liu et al., 2019; Chen et al., 2020a). Along this direction, instead of building a fully connected graph, recent works utilize auxiliary task structure/knowledge graph to build the graph (Chen et al., 2019; Lee et al., 2018). In contrast, and orthogonal to the papers above, we focus on modeling data-task relationships with the reinterpretation of the last layer. In addition, with data-task relationships solely, we are still able to capture data-data, task-task relationships implicitly through higher-order message passing.

## 6  CONCLUSION

We introduced relational multi-task settings in which the methods are required to learn to leverage labels on auxiliary tasks to predict on the new task. These settings are impactful in the biomedical domain where labels of different tasks are often scarcely available. To address these settings, we propose MetaLink, which is general enough to allow us formulating the above settings in a single framework. We demonstrated that MetaLink can successfully utilize the relations among tasks, outperforming the state-of-the-art methods under the proposed relational multi-task learning setting, with up to 27% improvement in ROC AUC. We limit our focus to model data-task relationships since most benchmark datasets do not have information on data-data or task-task relationships, though MetaLink is expressive enough to model such relationships. We leave extending MetaLink to more complex relationships or tasks for future work.

## ACKNOWLEDGEMENT

We want to give special thanks to Hamed Nilforoshan, Michihiro Yasunaga, Weihua Hu, Paridhi Maheshwari, Xuechen Li, Wengong Jin, Octavian Ganea, Tommi Jaakkola, and Regina Barzilay for the thoughtful discussions. We gratefully acknowledge the support of DARPA under Nos. HR00112190039 (TAMI), N660011924033 (MCS); ARO under Nos. W911NF-16-1-0342 (MURI), W911NF-16-1-0171 (DURIP); NSF under Nos. OAC-1835598 (CINES), OAC-1934578 (HDR), CCF-1918940 (Expeditions), IIS-2030477 (RAPID), NIH under No. R56LM013365; Stanford Data Science Initiative, Wu Tsai Neurosciences Institute, Chan Zuckerberg Biohub, Amazon, JPMorgan Chase, Docomo, Hitachi, Intel, KDDI, Toshiba, NEC, and UnitedHealth Group. Jiaxuan You is supported by JPMC PhD Fellowship and Baidu Scholarship.

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

## A    ADDITIONAL IMPLEMENTATION DETAILS

**Implementation details for biomedical datasets.** We use an adapted version of GraphSAGE Hamilton et al. (2017) as the base neural architecture, and implement MetaLink based on the descriptions in the main paper. We use Adam optimizer, with initial learning of $0.001$ and cosine learning rate scheduler. The model is trained with a batch size of 128 for 50 epochs. We search over the number of layers of $[2, 3, 4, 5]$, and report the test set performance when the best validation set performance is reached.

**Implementation details for MS-COCO.** We use ResNet-50 with an input size of $224 \times 224$ as the base neural architecture, and implement MetaLink based on the descriptions in the main paper. We perform standard data augmentation, including random crop and horizontal flipping. We use Adam optimizer with an initial learning rate of $0.002$, weight decay of $1 \times 10^{-4}$. The model is trained with a batch size of 256 for 40 epochs. We use the same set of hyper-parameters for all the four settings mentioned in the main paper.

**Implementation details for few-shot learning.** We use the widely adopted backbones ResNet-12 with an input size of $84 \times 84$ (Chen et al., 2020b). We perform standard data augmentation, including random crop and horizontal flipping. We use SGD optimizer with an initial learning rate of $0.001$, momentum of $0.9$, weight decay of $5 \times 10^{-4}$. The model is trained with a batch size of 4 for 20 epochs. Notice one batch includes a support set (with $5 \times 5$ examples) and a query set(with 15 examples). In a nutshell, we use the same set of hyperparameters as (Chen et al., 2020b), which is exactly the degenerated model if we set KG layer = 0.

**Additional algorithms.** We summarize the steps of MetaLink for relational setting in Algorithm 2, for meta setting in Algorithm 3.

**Code release.** We will make all the source code public at the time of publication.

---

**Algorithm 2** MetaLink Training in Relational Setting

---

**Require:** Dataset $\mathcal{D}_{\text{train}} = \{(\mathbf{x}, y)\}$. A parameterized embedding function $f_\theta$. Last layer weights for each task $\{\mathbf{w}_j\}$. A parameterized heterogeneous GNN $f_\phi$. Number of GNN layers $L$.

1: **for** each iteration **do**
2:      $\{(\mathbf{x}, \{y^{(i)}_{j \in T^{(i)}_{\text{aux}}}\}, \{y^{(i)}_{j \in T^{(i)}_{\text{test}}}\}\} \leftarrow \text{SampleMiniBatch}(\mathcal{D}_{\text{train}})$
3:      $\{\mathbf{z}\} \leftarrow f_\theta(\mathbf{x})$ for $\mathbf{x} \in \{(\mathbf{x}, \{y^{(i)}_{j \in T^{(i)}_{\text{aux}}}\}, \{y^{(i)}_{j \in T^{(i)}_{\text{test}}}\}\}$
4:      $V_d^{(0)} = \{\mathbf{h}_i^{(0)} \leftarrow \mathbf{z} \text{ for } \mathbf{z} \in \{\mathbf{z}\}\}$                                   ▷ Initialize data nodes
5:      $V_t^{(0)} = \{\mathbf{h}_j^{(0)} \leftarrow \mathbf{w}_j \text{ for } \mathbf{w}_j \in \{\mathbf{w}_j\}\}$                       ▷ Initialize task nodes
6:      $E = \{\mathbf{e}_{ij} \leftarrow (\mathbf{x}^{(i)}, t_j) \text{ for } y^{(i)}_j \in \{y^{(i)}_{j \in T_{\text{aux}}}\}$               ▷ Initialize edges
7:      **for** $l = 1$ to $L$ **do**
8:           $V_d^{(l)}, V_t^{(l)} \leftarrow \text{GraphConv}(V_d^{(l-1)}, V_t^{(l-1)}, E)$ with $f_\phi$
9:      $\text{logits} \leftarrow \text{EdgePred}(V_d^{(L)}, V_t^{(L)})$ with $f_\phi$
10:     $\text{Backward}\left(\text{Criterion}(\text{logits}, \{\{y^{(i)}_j\}_{j \in T^{(i)}_{\text{test}}})\right)$

---

## B    ADDITIONAL DISCUSSIONS

**Limitations.** i) In principle, MetaLink leverages task correlation to achieve data efficiency/gain better performance. Thus, it is less effective when tasks are not correlated. We believe it is fair to assume a multi-task learning system will work worse when tasks as not correlated. ii) Though our method is general to model diverse multi-task scenarios, we limit our empirical study to multi-label learning mostly due to the limitation of dataset availability. We hope that the proposed MetaLink can inspire researchers to collect more multi-task learning datasets with diverse use cases.

---

**Algorithm 3** MetaLink Training in Meta Setting

---

**Require:** Dataset $\mathcal{D}_{\text{train}} = \{(\mathbf{x}, y)\}$. A parameterized embedding function $f_\theta$. A parameterized heterogeneous GNN $f_\phi$. Number of GNN layers $L$.

1: **for** each iteration **do**
2:     $S, Q \leftarrow \text{SampleMiniBatch}(\mathcal{D}_{\text{train}})$                            $\triangleright$ Simulate meta setting in training
3:     $\{\mathbf{z}\} \leftarrow f_\theta(\mathbf{x})$ for $\mathbf{x} \in (S, Q)$
4:     $V_d^{(0)} = \{\mathbf{h}_i^{(0)} \leftarrow \mathbf{z} \text{ for } \mathbf{z} \in \{\mathbf{z}\}\}$                      $\triangleright$ Initialize data nodes
5:     $V_t^{(0)} = \{\mathbf{h}_j^{(0)} \leftarrow \mathbf{1}\}$                                  $\triangleright$ Initialize task nodes
6:     $E = \{\mathbf{e}_{ij} \leftarrow (\mathbf{x}^{(i)}, t_j) \text{ for } y_j^{(i)} \in S\}$             $\triangleright$ Initialize edges
7:     **for** $l = 1$ to $L$ **do**
8:         $V_d^{(l)}, V_t^{(l)} \leftarrow \text{GraphConv}(V_d^{(l-1)}, V_t^{(l-1)}, E)$ with $f_\phi$
9:     $\text{logits} \leftarrow \text{EdgePred}(V_d^{(L)}, V_t^{(L)})$ with $f_\phi$
10:    $\text{Backward}\left(\text{Criterion}(\text{logits}, \{\{y_j^{(i)}\}_{j \in T_s}\} \in Q)\right)$

---

**Social impact.** In practice, collecting high-quality datasets is often expensive and time-consuming. In the biomedical domain, labeling requires domain expertise and hence is very resource-intensive. In some applications, it requires long-term experiments to get the ground truth labels. The approach that we proposed could reduce the high labeling cost by utilizing auxiliary task labels that are already available. In principle, MetaLink attempts to leverage correlation among tasks. Thus, when it comes to future deployment, we suggest being careful about defining tasks. Because it might come across similar issues related to fairness as in other supervised learning problems, where the algorithm learns biased correlations from the datasets that are not considered to be appropriate. The datasets we used for experiments are among the most widely-used benchmarks, which should not contain any undesirable bias.

## C  ADDITIONAL RESULTS

**Advantages on using heterogeneous weights.** As discussed in Section 3, we proposed to learn from the knowledge graph constructed via a heterogeneous GNN. Here we empirically demonstrate the improvement of using distinct weights for updating data nodes and task nodes. Results are summarized in Table 5.

Table 5: Results of ROC AUC (for biomedical) and mAP (for MS-COCO) datasets.

| Setting | Weights | Tox21 (12 tasks) | Sider (27 tasks) | ToxCast (617 tasks) | MS-COCO (80 tasks) |
|---------|---------|------------------|------------------|---------------------|--------------------|
| Relational | Shared | $83.1 \pm 1.6$ | $75.8 \pm 2.5$ | $78.1 \pm 1.3$ | $74.02 \pm 0.16$ |
| | Hetero | $\mathbf{83.7 \pm 1.9}$ | $\mathbf{76.8 \pm 3.0}$ | $\mathbf{79.4 \pm 1.0}$ | $\mathbf{75.36 \pm 0.16}$ |

