# OpenReview forum: "Relational Multi-Task Learning: Modeling Relations between Data and Tasks"
_ICLR.cc/2022/Conference — ICLR 2022 Spotlight_

### Official Review · Reviewer_ijJF · 2021-11-02

**Correctness:** 4
**Technical Novelty And Significance:** 3
**Empirical Novelty And Significance:** 3
**Recommendation:** 8
**Confidence:** 4

**Main Review:**

Strengths:
- The paper is very well-organized and well-written. Algorithm 1 is self-explanatory.

- I really liked the idea of creating a knowledge graph and then solve the link label prediction problem.

- The experiments are thorough, I enjoyed reading the experiments section; they cover ablation studies, and answer several question in there.  Specifically how relational ML could take advantage of more related tasks. In addition, the improvements are noticeable (Tables 1-4).


Notes:
- In figure 1, in the right panel, I would put 1's and 0's in a right place, e.g., 1's for the solid lines or 0's with the dashed line?!

- I know that I have seen figure 2 somewhere before, but do not remember the paper, if you have seen this figure from another paper, that will be a good idea to put the reference.

- I would recommend defining use the name of your model in figure 4 instead of using 'ours'

**Summary Of The Paper:**

In this paper, they introduce relational multi-task learning in which they construct a graph with the data points and task and labels as edges. They represent the data points as the embedding of NN model and task as the last layer of NN model for that task. Then they solve the link label prediction problem between each node and task by using GNN and heterogeneous message passing method to predict the label of test data point on a new task at inference time.

**Summary Of The Review:**

In general, I believe the paper is innovative and descent. The backed their proposal with thorough experiments. I think this work will be useful for researchers working in the area of multi-tasking.

---

> ### Author Response · Authors · 2021-11-16
> **Response to Reviewer ijJF**
>
> We thank the reviewer for the valuable feedback and insightful comments. We sincerely appreciate the reviewer for confirming that our paper is well-written, innovative and thorough. We would like to make a few clarifications according to the reviewer’s comments as below.
>
> **Q1: In figure 1, in the right panel, I would put 1's and 0's in a right place, e.g., 1's for the solid lines or 0's with the dashed line?**
>
> We thank the reviewer for the suggestion. We would like to clarify that we use solid lines to represent known labels (which can be 0, 1, or others) and dashed lines for unknown labels. We will follow your suggestion and improve the aesthetics of Figure 1 in the final version.
>
> **Q2: If you have seen figure 2 from another paper, that will be a good idea to put the reference.**
>
> We thank the reviewer for the suggestion. We independently created this figure without referring to existing literature. The plot appears familiar to the reviewer perhaps because we used the same software. If needed, we appreciate it if the reviewer can point us to the relevant paper, and we will make sure to properly cite it.
>
> **Q3: I would recommend defining use the name of your model in figure 4 instead of using 'ours'.**
>
> Thanks for the suggestion. We have updated the name in the revised draft accordingly.

---

### Official Review · Reviewer_2M5P · 2021-11-02

**Correctness:** 4
**Technical Novelty And Significance:** 3
**Empirical Novelty And Significance:** 4
**Recommendation:** 6
**Confidence:** 4

**Main Review:**

Strengths
- Innovative formulation
-- a wide “range of heterogeneity” in the graph: a node for each sample in the support set a node for each task - backed up by relevant experimental improvements
-- dynamical graph structure based on the existing labels for the current entry
- Experimental results are strong for the few-shot setup and medical datasets
- Ablations are very informative: higher improvement in more correlated tasks and using more auxiliary tasks come with higher performance

Weaknesses
- Table 1. Why are there so many variations of the KG layer (both inside the same dataset and per metric)? Is there some consistent explanation for this hyperparameter?
- Even though it is quite simplistic, Figure 2 is confusing. The authors should highlight somehow that the edges show the conceptual connections between nodes (and not the real link in the algorithm flow). Such an example: in c) and d) why are data nodes (the ones from the triangle structures) useful if they are already connected via known label edges?
- Ablations on the graph structure decision would be useful (the ones from eq. 2 and 3, sustaining the “special GNN design”)
- Prior work is focused on GNN approaches (even in the multi-task paragraph), comparison from the design point of view with prior-art approaches in relational multi-task graphs could offer a more complete view (eg. Robust Learning Through Cross-Task Consistency - https://arxiv.org/abs/2006.04096, Self-Supervised Learning in Multi-Task Graphs through Iterative Consensus Shift - https://arxiv.org/abs/2103.14417)

Questions
- How many parameters does MetaLink learn?
- How could the authors adapt the proposed MetaLink to work with dense predictions rather than 1-D tasks? What would be the implications?

Others (minor, eg. typos)
- KG layer (knowledge graph) is never expanded
- Please mention what you show in the tables (top5 performance)
- “where T = {1, 2, ..., m} is integers between 1 and m.”
- Eq. 1 should have an (l-1) instead of (l) - it uses only h(l)


**Summary Of The Paper:**

The paper exploits the multi-task modeling using a heterogenous Graph Neural Network. The key contribution of the work is having both data (sample) and task nodes in the same graph, focusing on data-task relation, accommodating for sparse task labels by design.

Experimental results show relevant improvements with the proposed relational model on biomedical datasets and Imagenet splits for few-shot.


**Summary Of The Review:**

The paper is sound, introducing and validating an interesting heterogeneous graph design focused on each task and data sample as a different node. The work also offers an approach for the common labeling limitation in multi-task problems (by working with sparse labels in a dynamic graph architecture ). I recommend acceptance.

---

> ### Author Response · Authors · 2021-11-16
> **Response to Reviewer 2M5P**
>
> We thank the reviewer for valuable feedback and insightful suggestions. We also appreciate the reviewer for thinking this paper innovative, and the results are informative and strong. We hope the newly provided clarification could help to further strengthen our work.
>
> **Q1: Table 1. Why are there so many variations of the KG layer?**
>
> We thank the reviewer for the question. We select the KG layer with the best performance on the validation set. Our purpose is to facilitate the reproducibility of our paper. Besides, we observe that when the number of tasks is small, e.g. Tox21 with 12 tasks, the optimal number of KG layer is small; whereas when the number of tasks is large, e.g., ToxCast with 617 tasks, the optimal number of KG layers is large. Our explanation is that the optimal number of KG layers indicates how useful it is to learn from data-task interaction. Such information is especially important when the number of tasks is large, in which case a given task can significantly benefit from the information from the large pool of auxiliary tasks. We hope the variability of KG layers justifies the effectiveness of MetaLink.
>
> **Q2: Figure 2 is confusing. Why are data nodes (the ones from the triangle structures) useful if they are already connected via known label edges?**
>
> We thank the reviewer for the question. The data nodes (the ones from the triangle structures) are useful, because: (1) They are used to compute the task embeddings, and accurate task embeddings are especially important in the meta-learning settings where the few-shot data points are provided to a given task. (2) Their labels may serve as auxiliary task labels to another task; by propagation interaction between data and task nodes, MetaLink can use such auxiliary information to boost the performance of other tasks.
>
> **Q3: Ablations on the graph structure decision would be useful.**
>
> We thank the reviewer for the suggestion. We have included an ablation study on whether to use a heterogeneous graph representation or simply using a homogenous graph representation in Appendix C. We will include studies on other popular model architectures in the final version.
>
> **Q4: Comparison from the design point of view with prior-art approaches in relational multi-task graphs could offer a more complete view.**
>
> We appreciate the reviewer for referring us to the two related works. These works study the potential benefits of enforcing prediction consistency among tasks, and their setting is the same as the vanilla multi-task setting. We believe this line of work is orthogonal and complementary to our relational setting. We have included this discussion in the revised version.
>
> **Q5: How many parameters does MetaLink learn?**
>
> We thank the reviewer for the question. The number of parameters should depend on number of KG layer $l$ and the dimension of feature $h$. It should lie with $O(l \cdot h^2)$. In Table 1, when the number of KG layers equals 5, MetaLink has approximately 20M parameters.
>
> **Q6: How could the authors adapt the proposed MetaLink to work with dense predictions rather than 1-D tasks? What would be the implications?**
>
> We thank the reviewer for the question. We believe there’s no conceptual difficulty for extending MetaLink to dense predictions. The concepts of data nodes and task nodes still hold even if the task is more complex. Besides, we can extend the edge features from a scalar to a vector or tensor to encode more complex task labels. Nevertheless, on the empirical side, we acknowledge that it is nontrivial to implement such extensions, and we hereby leave it for future work. We hope the 4 multi-task datasets in biomedical and computer vision use cases (with up to 617 tasks) can justify the effectiveness of MetaLink.

---

> > ### Comment · Reviewer_2M5P · 2021-12-03
> > **Response to authors**
> >
> > Thank the authors for the responses. I keep my original rating and recommend acceptance.

---

### Official Review · Reviewer_jGkV · 2021-11-02

**Correctness:** 4
**Technical Novelty And Significance:** 3
**Empirical Novelty And Significance:** 3
**Recommendation:** 6
**Confidence:** 4

**Main Review:**

Strengths:
1. MetaLink is able to leverage the correlations among tasks successfully.
2. I find the idea of creating the knowledge graph for harnessing the relational information about the data and tasks very intreging.
3. The usage of the labels of the auxiliary task is well motivated.

Weakness:
1. The writing could have been more precise in some parts of the paper.

**Summary Of The Paper:**

The paper introduces a novel multi-task learning framework called MetaLink that takes the opportunity to utilize the auxiliary task labels by constructing a knowledge base with the tasks and data(instances) as nodes and auxiliary labels as the edge labels between them. This method shows significant empirical success

**Summary Of The Review:**

Concerns:
1. The introduction could have been more concise, I feel the authors could have avoided explaining the technicalities of the knowledge graph in the introduction. Rather, it would help the reader to be better situated in your research space if you discuss the fundamental difference between some of the related models and yours at an early stage of the paper. For example, the authors could have mentioned Wang et al. 2016 and ML-GCN  (Chen et al.) in the introduction itself. and explained why their work is different from the others.

2. Are the comparisons fair? The GraphConv( ) step uses additional parameters, especially when the number of layers (iteration) increases. Could you include the number of parameters for all the baselines and your method?

3. The tasks are very specific to multilabel classification. What happens when objectives are different for each task? How would MetalLink adapt to that?

minor: In the paragraph: "Results on biochemical datasets.", Table 5 ---> Table 1.

---

> ### Author Response · Authors · 2021-11-16
> **Response to Reviewer jGkV**
>
> We thank the reviewer for the suggestion and positive evaluation of our work. We are glad to hear that the reviewer appreciates that the idea is intriguing and well-motivated. We would like to make a few clarifications according to the reviewer’s comments below. We hope the newly provided clarification could help to further strengthen our work.
>
> **Q1: The introduction could have been more concise. It would help the reader to be better situated in your research space if you discuss the fundamental difference between some of the related models.**
>
> We thank the reviewer for the suggestion. We have included some discussions in the introduction following the comment. More thorough comparisons can also be found in Section 5.
>
> **Q2: The GraphConv( ) step uses additional parameters, especially when the number of layers (iteration) increases. Could you include the number of parameters for all the baselines and your method?**
>
> We thank the reviewer for the suggestion. The results of the baselines are referred from the original paper, therefore we are unable to get the exact number of parameters for them. In the meantime, in Table 1, MetaLink uses less than 20M parameters, while the SOTA baseline GROVER uses up to 100M parameters.
> Additionally, we have ensured that the comparison across models is fair. For example in Table 1, in all the MetaLink experiments, we tried the number of KG layers from 1 to 5, and report the number of KG layers with the best performance in the validation set. This ensures that the comparison is fair even some models have more parameters (due to having more layers).
>
> **Q3: The tasks are very specific to multilabel classification. What happens when objectives are different for each task? How would MetalLink adapt to that?**
>
> We thank the reviewer for the question. We believe there’s no conceptual difficulty for extending MetaLink to cases where the objectives are different for each task. The concepts of data nodes and task nodes still hold even if the task objectives are diverse. After conducting rounds of message passing among data and task nodes, we can train the data/task embeddings using the objective specific to the given task; when the number of GraphConv layers is 0, MetaLink reduces to a standard multi-task learning model. Nevertheless, on the empirical side, we acknowledge that it is nontrivial to implement such extensions, and we hereby leave it for future work.

---

### Official Review · Reviewer_QALx · 2021-11-05

**Correctness:** 3
**Technical Novelty And Significance:** 2
**Empirical Novelty And Significance:** 2
**Recommendation:** 5
**Confidence:** 4

**Main Review:**

According to the setting introduced in Section 2, it seems that the proposed setting is just a meta learning setting but not a multi-task learning setting. The name of relational multi-task learning seems a bit misleading.

In the first paragraph of Section 3.1, the claim is not general, since in many cases, the task head consists of multiple layers instead of only one layer. In the multi-layer case, based on the second paragraph of Section 3.2, the proposed MetaLink model seems not applicable.

The knowledge graph stated in this paper is just a graph. This is different from the actual ‘knowledge graph’ and this seems misleading.

The heterogeneous GNN used in Section 3.2 seems no difference with the conventional GNN. I did not see significant novelty in the proposed MetaLink model.

In experiments, only multi-label datasets are used. Authors should experiment on multi-task benchmark datasets such as CityScape, NYUv2, Pascal-Context, Taskonomy, Office-31, Office-Home.

**Summary Of The Paper:**

This paper proposes a relational multi-task learning setting and design a MetaLink model.

**Summary Of The Review:**

The proposed setting seems a bit misleading. The proposed model has no significant novelty. More standard benckmark datasets should be used.

---

> ### Author Response · Authors · 2021-11-16
> **Response to Reviewer QALx**
>
> We thank the reviewer for the valuable feedback and insightful comments. We respectfully ask the reviewer to consider increasing the score if our clarification has addressed the concerns raised by the reviewer. We have updated our paper to reflect the reviewer’s constructive suggestions.
>
> **Q1: It seems that the proposed setting is just a meta learning setting but not a multi-task learning setting. The name of relational multi-task learning seems a bit misleading.**
>
> We thank the reviewer for the comment and respectfully disagree.
> Finn et al. [1] nicely summarize the goal of meta-learning as “solve new learning tasks using only a small number of training samples”. In Section 2, we discuss a more generic multi-task learning setting, where the learning tasks are not necessarily new. In Figure 2, we show the variants of task settings that are considered: (1) relational setting, where sparse labels from auxiliary tasks can facilitate learning the tasks of interest; (2) meta-learning setting, where the tasks of interest are unknown at training time. Overall, the relational setting and meta-learning setting are two different cases that the proposed MetaLink framework can address.
>
> [1] Finn, Chelsea, Pieter Abbeel, and Sergey Levine. "Model-agnostic meta-learning for fast adaptation of deep networks." ICML 2017
>
> **Q2: The claim is not general, since in many cases, the task head consists of multiple layers instead of only one layer.**
>
> We thank the reviewer for the suggestion. Our MetaLink framework can generalize to multi-layer task heads. In Section 3.1, we assumed a task head is a single layer for ease of discussion; such a simplification does not lose the generality, as we can always view the last layer of a neural network as the task head. We further want to emphasize that MetaLink still enjoys an improved expressive power for multi-layer task head as well, since MetaLink introduces layers of message passing between data points and tasks before making the final (multi-layer) prediction.
> Following the reviewer’s feedback, we have added the discussion of multi-layer prediction head to Section 3.1.
>
> **Q3: The knowledge graph stated in this paper is just a graph. This is different from the actual ‘knowledge graph’ and this seems misleading.**
>
> We thank the reviewer for the suggestions. In the literature, “graph” refers to a homogenous graph, where nodes and edges do not have types. In MetaLink, we use the term “knowledge graph” to emphasize that the graph is heterogeneous (data nodes and task nodes), and can be represented as triplets (data point X, has_label, task Y). We also want to note that knowledge graph is a special type of heterogeneous graph, and they are often used interchangeably in the literature. We will add this clarification to better motivate why the term “knowledge graph” is used.
>
> **Q4: The heterogeneous GNN used in Section 3.2 seems no difference with the conventional GNN. I did not see significant novelty in the proposed MetaLink model.**
>
> We thank the reviewer for the comment. We would like to argue that our main contribution is not about the specific GNN architecture, but lies in (1) a generic definition of multi-task learning settings, and (2) jointly solving these different settings via a unified MetaLink framework, i.e., reducing multi-task learning into a link prediction problem on a data-task graph. Our MetaLink framework is general thus does not require specific GNN architecture to proceed. In Section 3.2, we have discussed the “Special GNN designs in MetaLink” as well.
>
> **Q5: In experiments, only multi-label datasets are used. Authors should experiment on multi-task benchmark datasets such as CityScape, NYUv2, Pascal-Context, Taskonomy, Office-31, Office-Home.**
>
> We thank the reviewer for the question. We want to note that multi-label datasets are good testbeds for multi-task learning tasks; due to various reasons, people often group the labels for multiple tasks into one dataset. For example, the Sider dataset that we used contains 27 unique tasks on adverse drug reactions.
> We believe there’s no conceptual difficulty for extending MetaLink to multi-task datasets. The concepts of data nodes and task nodes still hold even if the task heads are more complex (Please refer to our response in Q2). Besides, we can extend the edge features from a scalar to a vector to encode more complex task labels. Nevertheless, on the empirical side, we acknowledge that it is nontrivial to implement such extensions, and we hereby leave it for future work. We hope the 4 multi-task datasets in biomedical and computer vision use cases (with up to 617 tasks) can justify the effectiveness of MetaLink.

---

### Author Response · Authors · 2021-11-28
**Gentle reminder: author response period will end soon**

Dear Reviewers,

We sincerely appreciate your time and efforts in reviewing our paper.

We tried our best to reply to all the concerns and questions raised in the review. We hope the additional explanations and clarifications can help to improve your evaluation of our paper. We have made significant improvements thanks to the comments from all the reviewers.

Please feel free to let us know if you have any outstanding questions or concerns. Thank you very much!

---

### Decision · Program_Chairs · 2022-01-20

**Decision:**

Accept (Spotlight)

**Comment:**

The paper describes a novel learning scenario where there are many related tasks, some seen at test time, and some seen only at training time, where additionally the task labels can be hidden or present.  This approach generalizes both a "relational setting" (where auxiliary task labels could be used as features) and a "meta setting" (where new tasks need to be solved in a zero-shot setting using data from related tasks only).  The idea behind the method is to do MTL with a common representation and a set of task-specific heads, and build a graph where (1) tasks are nodes associated with the parameters of their task-specific "heads" and (2) edges link examples to tasks with known labels.  A GNN method is then used to find regularities in the graph.

Pros
 - The setting is innovative and the approach is novel
 - The experimental results are strong

Cons
 - Some of the terminology seems awkward and/or strained (eg "knowledge graph" for the task-example graph)